# The Asymmetric and Symmetric Effect of Energy Productivity on Environmental Quality in the Era of Industry 4.0: Empirical Evidence from Portugal

James Karmoh Sowah, Jr. [1], Sema Yilmaz Genc [2], Rui Alexandre Castanho [3,4], Gualter Couto [5], Mehmet Altuntas [6] and Dervis Kirikkaleli [7,*]

1   Department of Business Administration, Faculty of Economic and Administrative Sciences, European University of Lefke, Lefke 99728, Northern Cyprus, Turkey
2   Department of Economics, Yildiz Technical University, Istanbul 34349, Turkey
3   Faculty of Applied Sciences, WSB University, 41-300 Dąbrowa Górnicza, Poland
4   College of Business and Economics, University of Johannesburg, P.O. Box 524, Auckland Park, Johannesburg 2006, South Africa
5   School of Business and Economics and CEEAplA, University of Azores, 9500-321 Ponta Delgada, Portugal
6   Department of Economics, Faculty of Economics, Administrative and Social Sciences, Nisantasi University, Istanbul 34000, Turkey
7   Department of Banking and Finance, Faculty of Economics and Administrative Sciences, European University of Lefke, Lefke 99728, Northern Cyprus, Turkey
*   Correspondence: dkirikkaleli@eul.edu.tr

**Abstract:** Energy has never been used in the same way or to the extent that it is today. The $CO_2$ level in the atmosphere surpassed the previous record established in 1958 in May 2019 when it hit 415.26 ppm, and the climate system has reached a tipping point. New corporate initiatives are required to create more sustainable eco-market opportunities and enhance stewardship in order to make the transition to net zero carbon emissions. This research investigates the asymmetric and symmetric impact of energy efficiency on environmental quality in Portugal from 1990Q1 to 2020Q4, while accounting for the role of total energy consumption (TEC), trade openness (TRA), and economic growth (GDP) in driving environmental quality in the era of industry 4.0. Portugal has emerged as a crucial player, experiencing rapid economic and financial growth, and attracting an unprecedented inflow of foreign trade. While country growth is appreciable in the monetary sense, this research employs the nonlinear autoregressive distributive lag (NARDL) technique and econometric robustness tests to examine the consequence of $CO_2$ emissions in Portugal. The results verify the asymmetric (different magnitude) impacts across the modeled variables; specifically, a 1% volatility to energy productivity (EP) reduces environmental degradation in Portugal by 3.247606%, while a 1% volatility to GDP, TRA, and TEC increase environmental degradation in Portugal by 0.29119%, 0.717775%, and 0.034088% over the long-term. Energy productivity sources are a great way to help Portugal keep its energy independence and reduce environmental erosion simultaneously. Switching from nonrenewable energy to investing in low-carbon technology is a crucial strategy for decarbonization and the best practical course of policy action for reducing climate change in Portugal.

**Keywords:** energy productivity; environment; asymmetric; NARDL; symmetric; Portugal





## 1. Introduction

Energy has traditionally been a necessary element of human existence and economic output, supporting overall socio-economic growth. The hidden force that has propelled humanity's incredible advancement is the deployment of new energy sources. Because of rising economic activity and population growth, there is a greater demand for energy globally [1,2]. The current energy scenario predicts that by 2030, the world's energy consumption will have at least doubled and that by 2050, it may have tripled to 722

quadrillions Btu [3]. More than 80% of the total energy consumed in most nations comes from fossil fuels. When fossil fuels are used, they leave behind solid and gaseous waste that causes environmental damage that cannot be undone [1,4]. The climatic system and biosphere are at a standstill. The year 2020 saw a new record for the amount of carbon dioxide in the atmosphere, surpassing the previous mark from 1958, which was 415.26 ppm. If comprehensive action is not taken, the amount of carbon dioxide being emitted from the world will keep increasing and stay in the atmosphere for a very long time [5,6], which will cause the sea level to rise. More extreme weather occurs frequently, and other severe environmental issues are increasing [7].

With the backdrop of climate-smart green investments, trade in renewable energy resources has advanced recently in developing and developed economies, particularly in developed nations. It has advanced relative to the access to climate mitigation strategies [8,9]. Renewable energy resources have shown great potential for generating heat and electricity. From the global perspective, the amount of electricity generated by renewable sources has gradually increased during the past 20 years (Figure 1). In 2021, more than half of the increase in the world's electricity supply came from renewable energy sources. Despite a breakdown in the supply chain and high raw material costs, the capacity of renewable energy generation surpassed 295 GW in 2021, reaching its most significant peak since 2000.

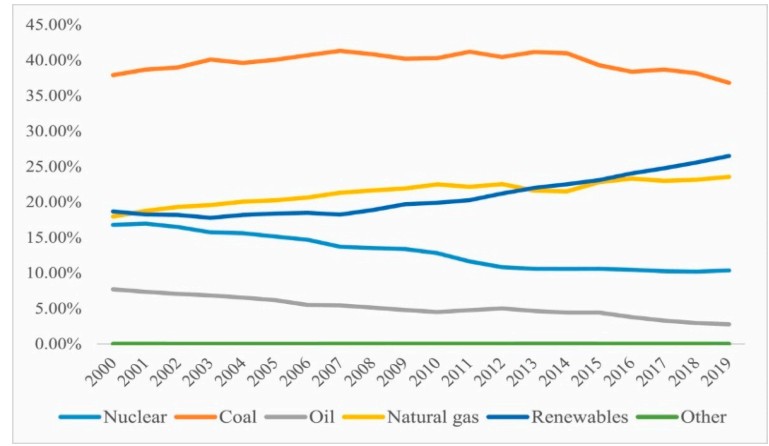

**Figure 1.** Visual overview of consistent development of the world's energy resources. Source: OECD.

In OECD nations, wind and solar PV spur are the primary renewable energy sources for generating electricity and have rapidly grown (Figure 2). Additionally, renewable energy helps provide energy security. On the other hand, geopolitics and other relevant factors significantly influence the supply of fossil fuels such as oil and natural gas. Regarding local renewable energy, there are no such restrictions. It has a more reliable energy supply and is innovatively orienteered. Given the advantages of renewable energy, numerous nations and global organizations have taken action to promote and expand the use of renewable energy in order to enhance environmental quality and guarantee energy security.

In particular, Germany, Canada, Australia, and France have all implemented tax reduction programs and subsidies to assist the renewable growth technologies and innovations sector. Over half of the increase in renewable resources generated globally has come from China's competitive renewable energy sector. More than fifty nations, including the entire EU, demanded that net zero emissions targets be realized through renewable energy development at the UN Climate Change Conference in Glasgow (COP 26). Figures 1 and 2 show a visual overview of the development of the world energy resource from 2000 to 2020.

The United Nations (UN) has developed several framework conventions on climate change in order to address environmental challenges and accelerate progress. The 2015 Paris Agreement was structured to bring nations together in order to address the threats posed by global warming and climate change in a proactive manner. For instance, the global average temperature rise should not exceed 2.0 degrees Celsius, and governments and

businesses should work to limit the rise to 1.5 degrees Celsius. As a result, the International Energy Agency (IEA) has stated that increasing the energy efficiency policy could help achieve the net zero emissions targets [10]. Furthermore, countries worldwide gathered in Sharm el-Sheikh, Egypt, from 6 November to 20 November 2022, to take action toward achieving the world's collective climate goals, as agreed under the Paris Agreement and the Convention.

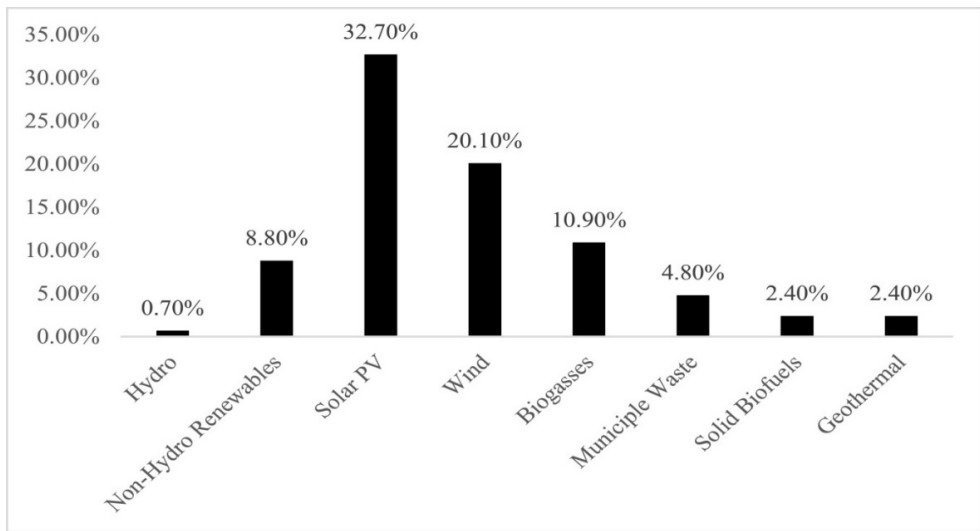

**Figure 2.** Visual overview of the world's renewable energy resource. Source: OECD.

Despite these efforts, climate threats remain complex; economists have conflicting opinions regarding which factors are effective in keeping global temperatures below 1.5 degrees Celsius [11]. With the deterioration of the environment worldwide, the connection between energy productivity and $CO_2$ has sparked much discussion. Exploring whether energy productivity can help reduce $CO_2$ emissions is crucial in order to design environmental policies. Hence, this study aims to explore the "asymmetric and symmetric" effects of energy efficiency on environmental quality in Portugal from 1990Q1 to 2020Q4, while accounting for the role of total energy consumption (TEC), trade openness (TRA), and economic growth (GDP) in driving environmental quality.

Portugal is a stunning nation that is committed to environmental and human rights protection. Portugal has grown tremendously since 1970 and is a signatory to the Kyoto Protocol and the UNFCC. Portugal has taken the lead in obtaining the right to a healthy environment that is recognized by the law. The country has also consistently backed historic resolutions in support of the human right to a clean, healthy, and sustainable environment on a global scale at the United Nations (UN) Human Rights Council [2021] and UN General Assembly [2022]. Despite these positive improvements, heatwaves, wildfires, and droughts seriously harm the nation's various ecosystems.

Over 1000 people have died due to heatwaves, and nearly 110,000 hectares of forest burned in 2022. The shift to climate neutrality is tough in several Portuguese regions, particularly in Alentejo, Litoral, and Medio Tejo, where the two respective coal-fired power facilities in the municipalities of Sines and Abrantes were shut down in 2021; further areas that need help transitioning to climate neutrality include In Sines and Matosinhos, where the production of highly polluting plastics and refined petroleum is practiced (in the Metropolitan Area of Porto). Across all industries, these factories produced the most greenhouse gases in Portugal. The intensity of Portugal's greenhouse gas (GHG) emissions, relative to its gross value added, is still about 30% higher than that of the European Union, where the major GHG emitters are the energy and transportation sectors.

Economic success, well-being, and convergence depend on productivity growth over time. Effective competition, business-friendly environments, and a fair and well-

functioning single market where small and medium-sized firms (SMFs) can operate and innovate without trouble is a significant tool for expanding economic productivity [12].

In contrast, energy productivity growth in Portugal has remained moderate, hampered, in particular, by low levels of investment, moderate levels of innovation capacity, and an overall skill level below the EU average; in addition, other business environment factors, such as undercapitalized firms and regulatory restrictions on the markets, are stifling competition in Portugal. Despite increased corporate innovation support over the past ten years, Portugal's innovation performance has yet to improve dramatically. Portugal saw slower GDP per capita growth than the rest of the EU between 2010 and 2019. In 2021, Portugal's GDP per capita was 79% of the EU average. Lisbon had the most significant GDP per capita among the Portuguese regions and was the only region with a higher GDP than the EU average. The Algarve and Madeira had a GDP per capita that was most comparable to the EU average (88% and 76%, respectively). The total support from the public for R&D investments has more than doubled over the past ten years, growing from 0.128% in 2010 to 0.264% in 2021, partly due to an expansive R&D tax incentive scheme. Businesses also raised the amount they spent on R&D, from 0.71% of GDP in 2010 to 0.92% in 2022.

In 2020, imported fossil fuels accounted for 69% of Portugal's gross inland energy consumption (42% from oil, 24% from natural gas, and 3% from coal). All fuels, including natural gas and coal, are imported. The NECP aims to reduce the energy import dependency to 65% by 2030, while achieving 80% renewable electricity production. Portugal is advancing, with plans to speed up the adoption of renewable energy, mainly solar photovoltaics, and finish up new hydroelectric projects. Portugal surpassed its objective of achieving a 31% gross final energy consumption in 2021 by achieving a 34% share of renewable energy. In 2022, 60% of its total electricity was produced using renewable energy sources. This included hydropower and wind generation (23.2% and 3.2% of the total electricity produced). However, since 2014, gas consumption has sharply increased due to rising power plant demand, partly due to a decrease in the amount of hydropower available due to persistent droughts. Due to pressure from the market and the law, Portugal's two coal-fired power plants will be shut down by private operators in 2023. The government estimates that natural gas-powered electricity production may continue until 2040. A summary of Portugal's energy use from 2005 to 2020 is shown in Figure 3.

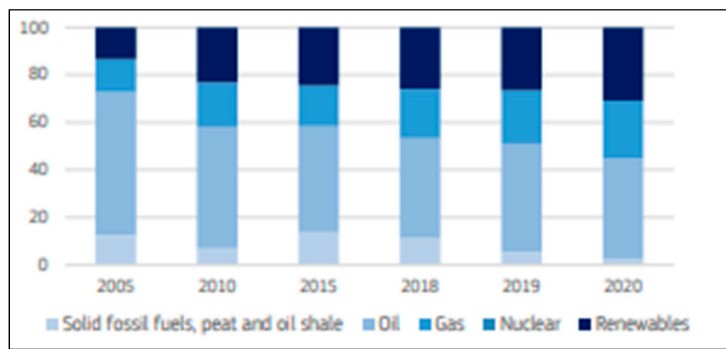

**Figure 3.** A pictorial summary of energy consumption in Portugal.

Based on the previous discourse, there is an urgent need to address Portugal's triple environmental catastrophe, which needs all stakeholders to acknowledge and rigorously respect, defend, and fulfill the human right to a healthy environment. This study contributes to the existing literature by exploring the asymmetric and symmetric effect of energy productivity on environmental quality in the era of industry 4.0, in particular, by providing empirical evidence from Portugal. However, the question is, are the two impacts of the same magnitude (symmetric effect)?

Most research implies that energy productivity (EP) reduces $CO_2$ emissions and environmental deterioration [5,13,14]. Energy productivity also helps restructure the economy from a factor-based model to an innovation-based one [15]. It minimizes publication bias

for Portugal, critically examining the function of TRA, TEC, and GDP growth as a driver of environmental quality. According to Wang [13] EP helps provide ecologically friendly energy, lessen enterprises' reliance on conventional energy, and foster green economic growth. Second, current papers on the consumption of fossil fuels and clean energy research remain scarce and controversial. They have been criticized for having Type I and Type II publication bias. Utilizing numerous robust scientific econometric approaches, such as [16]'s unit root test with a structural break, BDS nonlinear dependency test, nonlinear ARDL (NARDL) bounds test estimators, and other empirical robustness tests of canonical cointegration regression (CCR), dynamic OLS (DOLS) and fully modified OLS (FMOLS) in order to address the existing methodological gap in the literature in the context of Portugal; this was by providing an answer to the question of whether the impacts of the estimated variables have the same magnitude (symmetric effect) or a different magnitude (asymmetric). To the authors' knowledge, this is the first research to employ these resilient novelty econometric tools to shed new light on the determinants that have impacted investments in environmental quality in Portugal between 1990Q1 and 2021Q4. Given the current emphasis on climate-smart low-carbon investments, green growth, and addressing catastrophic losses caused by climate change, this study's findings provide a policy for achieving SDG 7, i.e., ensuring that everyone has access to modern, affordable, reliable, and sustainable energy, and SDG 13, i.e., taking immediate action to combat climate change and its consequences.

Additionally, this study considers these general characteristics and postulates the presence of interrelated nonlinearities or asymmetries with important policy ramifications. In a nutshell, the influence of energy productivity on carbon reduction is debatable, implying that the causal relationship between the two variables should be investigated further. This study's remaining sections are organized as follows: The relevant literature is reviewed in Section 2; the study's empirical model development is explained in Section 3; the model's estimations approach is presented in Section 4; the empirical findings and discussions are summarized in Section 5; and the conclusion, policy recommendations, study limitations, and some research ideas are presented in Section 6.

## 2. Review of the Related Literature

This study examines whether energy productivity's asymmetric or symmetric effect affects environmental quality in Portugal, while controlling for total energy consumption, trade openness, and economic growth. Debates about energy efficiency and market liberalization arose in EU politics in the 2010s, and they have become an essential part of EU energy policy since 2018 [17]. The use of energy efficiency applications is a comprehensive concept that aims to maximize the use of electricity, hydrogen power, solar energy, and other forms of energy for heating [18]. Energy efficiency has a variety of monetary costs that are borne by private and corporate actors in terms of benefits and costs. Its investments and business practices steadily reduce network demand in specific grid regions, postponing the need for the transmission or distribution of network upgrades. In contrast to Germany, Finland, Austria, India, and Turkey, energy efficiency policy in the United States is driven by public–private partnerships and state regulation [19,20]. In the United States, the energy efficiency markets are vertically integrated and unbundled regulated [21].

Theoretically, energy efficiency and productivity are guiding principles for energy-related investment concepts and policy development in a market economy. Production and consumption are determined by decentralized decisions made by corporations and private enterprises [1]. The extant research needs to establish the specific economic motivation for energy efficiency. However, it assumes that there are well-functioning markets for smart and efficient investments in zero energy emissions. This might range from dictatorships to the concentration of markets. In this section, the authors examine and summarize the most recent empirical research pertinent to neoclassical and regulatory economics [15].

Energy productivity is a relatively recent concept that measures the monetary value of total primary energy usage. It is measured in dollars per megajoule and indicates a nation's

widespread use of primary energy (TPES). Recent studies, such as [22–25], investigated the impact of energy efficiency on carbon emissions. In particular, Wang [13] examined energy efficiency usage in OECD economies. The study established that energy efficiency usage reduced $CO_2$ emissions in OECD countries between 1990 and 2019.

Besides, ref. [26] evaluated the drivers of energy productivity in 39 countries between 1995 and 2019. According to this study, increasing sectoral energy efficiency was critical to improving economic stability in the 39 selected countries. Despite these findings, the studies of refs. [6,27] discovered a variety of associated obstacles, issues, or hurdles that affects energy productivity policy implications in many nations. Moreover, the studies of refs. [6,27] argued that green innovations and technological improvements should be accelerated to improve environmental quality. This study formulates the following hypothesis (1), based on the empirical literature cited above:

**Hypothesis 1 (H1):** *Energy productivity policy mitigates $CO_2$ emissions in Portugal.*

In general, satisfying enormous energy consumption demands directly impacts environmental quality. In actuality, Portugal produces its domestic electricity using renewable domestic energy sources such as water, wind, sun, and biofuels. Furthermore, the Portuguese government has always prioritized energy security because Portugal imports nuclear fuels, biofuels, and fossil fuels such as oil and natural gas. In Portugal, the energy system is divided into supply energy and consumption energy. The weather, the economy, and energy efficiency are all factors that can influence energy consumption. In Portugal, more solar PV cells are being installed. In 2020, approximately 65,819 solar systems with a total installed capacity of 1090 MW were installed in Portugal. Renewable energy consumption has increased significantly in relation to total energy consumption since 2011, reaching just under 56% in 2019. The increased use of biofuels, combined with an increase in wind power production, has been the primary cause of the increase over the last year. In 2020, Portugal exported 25 TWh of electricity, most of which went to Finland, Lithuania, Denmark, and Poland. Following EU law, Portugal has committed to converting 100% of its electricity to renewable sources by 2040. In line with the related studies, this current study believes that in Portugal energy efficiency is likely to reduce $CO_2$ emissions, i.e., $\vartheta_1 = \frac{\vartheta LCO_2E}{\vartheta LEP_{it}} < 0$. The findings of studies by [1,24,28] are in line with this hypothesis.

A nation can expand and develop quickly by connecting to the global value chain through global trade. Trade is measured as the total exports and imports as a proportion of a country's GDP or regional economy. Openness to trade (TRA is usually viewed as a critical factor influencing the rigor of environmental policy. More stringent environmental regulations may encourage firms to create green and energy-saving technology—research on the empirical consequences of trade openness on $CO_2$ emissions [29–32].

These studies are of the utmost significance. TRA fosters the country's economy through convergence impacts. Studies from [33–35] offer compelling convergence evidence on TRA from different perspectives in terms of different countries' contexts. Therefore, the current study formulates the second hypothesis (2).

**Hypothesis 2 (H2):** *Trade openness has a positive effect on $CO_2$ emissions in Portugal.*

In addition, it is critical to stress that the OECD nations are today progressing faster than they were a few years ago. The trade openness of OECD nations is higher than that of developing nations due to their more robust growth rates. The implication is that economic openness may eventually have a detrimental effect on environmental quality in Portugal, while it is unclear whether commercial trade causes loss or stringent policy. With a low of 43.49% in 1991 and a high of 86.185 in 2008, Portugal's trade value during the review period, on average, was 63.38%, and the most recent number for 2020 is 72.08%. With a current score of 78.3%, Portugal is ranked sixth in Europe out of 45 countries and ninth in the 2022 index. Nevertheless, according to data from 2021, the COVID-19 rise significantly

negatively influenced Portugal, since commerce only made up 1.2% of the country's growth. The current analysis makes the assumption that trade openness has empirically increased $CO_2$ emissions in Portugal based on these data; i.e., $\vartheta_2 = \frac{\vartheta LCO_2 E}{\vartheta LTRA_{it}} > 0$; the studies of [36,37] support this paper's assumption.

Energy is a critical strategic issue in both developing and developed countries. Total energy consumption has more than doubled in the last three decades. A country's total energy consumption measures its total energy demands. Total energy consumption is classified into three types: (1) energy losses during transformation (e.g., oil or gas to electricity); (2) energy consumption by the energy sector itself; and (3) energy distribution to end users for final consumption. In 2018, fossil fuels accounted for 90% of global total energy consumption (WTPEC), with oil accounting for 34.77%, natural gas accounting for 23.76%, and coal accounting for 29.36%; meanwhile, nuclear fuels accounted for 5.47% and hydroelectricity accounted for 6.63%. Recent empirical studies on the effects of primary energy consumption on $CO_2$ emissions include those of [19,20]. According to Umar et al. (2021)'s study on the United States' transportation sector, fossil fuel consumption increased $CO_2$ emissions significantly from 1981 to 2019. Similarly, the findings of [38]'s study were supported by [5], which reported upon panel research on African countries. Contrarily, ref. [39]'s study could have offered more meaningful evidence. Based on this evidence, we assumed that the primary energy consumption in Portugal had increased $CO_2$ emissions; thus, this study develops hypothesis (3).

**Hypothesis 3 (H3):** *Economic growth increases $CO_2$ emissions in Portugal.*

Economic growth forecasting is difficult and complex, requiring several assumptions; thus, most empirical findings on the growth process are mixed [40–44]. Some researchers argue that economic growth positively impacts people's lives, while others argue that growth leads to higher carbon emissions, which is extremely rare. Portugal's GDP growth rate was 2% in 2019, and it is predicted to jump to 5.7% in 2023; however, its GDP shrank by 0.8% in the first quarter of 2022, but it is expected to grow by about 1.9% by 2023. As a result, it is expected that Portugal's economic expansion will lead to an increase in $CO_2$ emissions, i.e., $\vartheta_4 = \frac{\vartheta LCO_2 E}{\vartheta LGDP_{it}} > 0$. Contrary to research by [45–47], this assumption supports [19]'s study. Based on the inclusive debates from the above literature, further research is needed to fully understand the asymmetry and symmetric impact of energy productivity on Portugal's environmental quality. In order to close this gap in the environmental economics literature, the authors used the newly developed NARDL bound test model and other reliable econometric methods to report new evidence on productivity–energy–trade growth relationships.

## 3. Data Sources and Study Methodology

### 3.1. Data Sources

This study examines environmental quality in Portugal from the first quarter of 1990 to the fourth quarter of 2020, while controlling for economic growth, trade openness, and total energy consumption. The information comes from databases kept by the Organization for Economic Cooperation and Development (OECD), and is updated four times a year (i.e., using Eviews 12 to transform the datasets). Furthermore, the quarterly dataset is generated from annual data using the Quadratic approach in the Eviews 12 software. This study's dependent variable is the environmental quality proxy by carbon dioxide emissions ($CO_2$ emissions in tons) [43]. The explanatory variables are economic growth (GDP), energy productivity (EP), total energy consumption (TEC), and trade openness (TRA). Theories and empirical evidence were used to select these variables [38,48]. Each of these variables has human-related activities, and it is believed that they increase $CO_2$ emission concentrations in the atmosphere [49,50]. GDP is measured in US dollars per capita in 2010, TRA is measured in both the export and import of goods and services, including private and public investment stock and market capitalization as a percentage of GDP, and TEC is

measured in the total gross inland energy consumption. The primary explanatory variable is EP, a relatively new concept that quantifies the smart and green environmental and economic benefits of end users' final energy. All data in this work are expressed in natural logarithm form to avoid scaling concerns [6,48]. The variable in descriptive statistics is summarized in Table 1, while Figure 4 depicts the analytical flowchart.

**Table 1.** Summary Descriptive Statistics *.

| Period 1990Q1–2019Q4 | Data Source OECD | | | |
|---|---|---|---|---|
| **Code** | **LCO$_2$** | **LEP** | **LGDP** | **LTEC** | **LTRA** |
| Mean | 4.722729 | 4.114320 | 22.66005 | 5.594359 | 10.80702 |
| Median | 4.707227 | 4.112219 | 22.70486 | 5.658330 | 10.83867 |
| Maximum | 4.819353 | 4.197604 | 22.79576 | 5.712535 | 11.05537 |
| Minimum | 4.611560 | 4.067508 | 22.42230 | 5.261382 | 10.52096 |
| Std. Dev. | 0.056407 | 0.030632 | 0.106331 | 0.132165 | 0.151277 |
| Skewness | 0.065400 | 0.284748 | −0.865774 | −1.236958 | −0.416283 |
| Kurtosis | 1.927575 | 2.114371 | 2.244457 | 3.071820 | 2.091481 |
| Jarque-Bera | 547.8366 | 477.2611 | 262.8565 | 648.9456 | 125.3614 |
| Probability | 0.365906 | 0.107908 | 1.300220 | 2.008769 | 2.631729 |

* Note: OECD: Organization for Economic Co-operation and Development

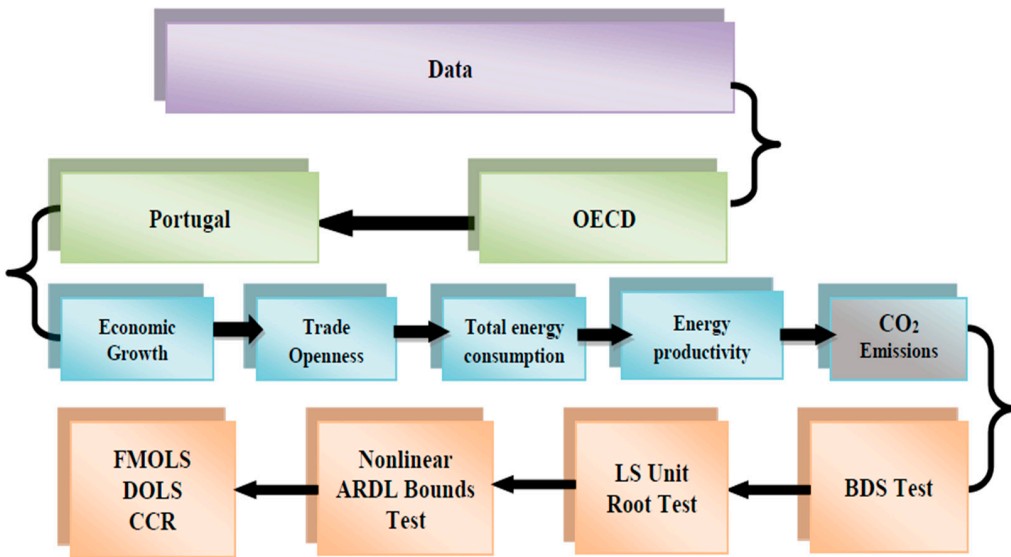

**Figure 4.** Analysis Flowchart. Source: Authors' computation.

*3.2. Empirical Model Development*

This study investigated the relationships between variables such as energy, economic growth, and the environment. It is crucial to note that energy productivity needs to receive more attention regarding its significance for $CO_2$ emission management. Similarly, the literature needs to include an empirical analysis of the Portuguese economy. The nonlinear ARDL estimator of [51] is expanded by adding energy productivity (EP) as a critical factor. The Equations (1)–(3) illustrate how variables are specified in the extended neoclassical growth model using the broad approach.

$$CO_2E_{it} = f(EP, TEC, GDP, TRA) \tag{1}$$

In order to avoid scaling problems, all variables are converted to natural logarithms in preparation for the empirical analysis.

$$LCO_2E_{it} = \vartheta_0 + \vartheta_1 LEP_{it} + \vartheta_2 LTEC_{it} + \vartheta_3 LGDP_{it} + \vartheta_4 LTRA_{it} + \varepsilon_{it} \tag{2}$$

where t indicates time and i represents linearity units. $CO_2E_{it}$ depicts carbon dioxide emissions, $EP_{it}$ denotes energy productivity, $TEC_{it}$ refers to total energy consumption, $GDP_{it}$ denotes economic growth, $TRA_{it}$ refers to trade openness, $\vartheta_0$ is the constant term and the standard error term is represented by $\varepsilon_{it}$. The use of both positive (POS) and negative (NEG) shock in the NARDL model is also presented as follows:

$$
\begin{aligned}
LCO_2E_{it} = \vartheta_0 + \vartheta_1 LEP_{1t}^+, LEP_{1t}^-, + \vartheta_2 LTEC_{2t}^+, LTEC_{2t}^- \\
+ \vartheta_3 LGDP_{3t}^+, LGDP_{3t}^- + \vartheta_4 LTRA_{4t}^+, LTRA_{4t}^- + \varepsilon_{it}
\end{aligned}
\tag{3}
$$

## 4. Models Estimations Approach

### 4.1. Unit Root Tests

This study investigates the connection between energy production and $CO_2$ emissions in Portugal by considering trade openness, total energy consumption, and economic growth. The LS structural break unit root tests of [16] were utilized to determine the integration properties of the time series variables. According to [52]'s study, most earlier studies have ignored structural breaks, leading to the stationarity of variables to favor false null hypothesis and bias; hence, [53]'s unit root is not allowed in this study. The mathematical model of [16] is presented as follows:

$$
Y_t = \delta' Z_t + \varepsilon_t, = \beta \varepsilon_{t-1} + e_t
\tag{4}
$$

where $Z_t$ refers to a vector of exogenous elements, and $\beta$ and $\delta'$ denote the hypothesis of the vector coefficients in the regression.

The study uses the LS model to ascertain whether the order of integration is linear or nonlinear dependent. According to [52], most earlier investigations neglected structural breaks, leading to unit root tests that were biased in favor of a false null hypothesis. Further, this current study employed [54]'s BDS test to detect the hidden stochastic dependency of the variables. It defends model misspecification and critical mistakes, which are its two main benefits. The economic applicability is as follows:

$$
BDS_{mT}(\varepsilon) = T^{1/2}[C_{m,T}(\varepsilon) - C_{1,T}(\varepsilon)^{\mathrm{m}}] / \delta_{mT}(\varepsilon)
\tag{5}
$$

where "$T$" denotes years covered by the study, $\varepsilon$ designates variables and $\delta_{mT}(\varepsilon)$ denotes the statistical numerator; meanwhile, the "$m$" presents the standard sample deviation that varies with each dimension.

### 4.2. Nonlinear ARDL Model Estimators

This study employs the nonlinear ARDL estimators to evaluate the long-term relationship among the selected variables. The results of this test are more reliable and robust. Comparing this model to other traditional estimators' techniques, the nonlinear ARDL offers various benefits: it can handle variable inappropriateness using the ECM model, and it allows a mixed integrated order as indicated:

$$
Y_t = \beta_o + \beta_1^+ X_t^+ + \beta_1^+ X_t^- + \mu_t
\tag{6}
$$

where $x_t^+$ and $x_t^-$ are the partial sums of changes $X_1$ that are both positive and negative. The long-run asymmetric and symmetric is computed by utilizing the following model: $L_{M1+} = \frac{-\varphi_1^+}{\rho}$ and $L_{M1-} = \frac{-\varphi_1^-}{\rho}$. This research is the structure in accordance with the NARDL model:

$$
\begin{aligned}
\Delta LCO_2E_t = \alpha_0 + \sum_{i=1}^{p} \alpha_1 \Delta CO_2E_{t-i} + \sum_{i=0}^{p} \alpha_3 \Delta LEP_{t-i}^+ + \sum_{i=0}^{q} \alpha_3 \Delta LEP_{t-i}^- + \\
\sum_{i=0}^{p} \alpha_2 \Delta LGDP_{t-i}^+ + \sum_{i=0}^{q} \alpha_3 \Delta LGDP_{t-i}^- + \sum_{i=0}^{p} \alpha_4 \Delta LTEC_{t-i}^+ + \sum_{i=0}^{q} \alpha_3 \Delta LTEC_{t-i}^- + \\
\sum_{i=0}^{p} \alpha_4 \Delta LTRA_{t-i}^+ \sum_{i=0}^{q} \alpha_3 \Delta LTRA_{t-i}^- + \rho LCO_2E_{t-1} + \varphi_1^+ LEP_{t-1}^+ + \varphi_2^- LEP_{t-1}^- + \\
\varphi_1^+ LGDP_{t-1}^+ + \varphi_2^- LGDP_{t-1}^- + \varphi_1^+ LTEC_{t-2}^+ + \varphi_2^- LTEC_{t-2}^- + \varphi_1^+ LTRA_{t-2}^+ + \varphi_2^- LTRA_{t-2}^- + \mu_t
\end{aligned}
\tag{7}
$$

where $\Delta Y_t = \alpha_0 + \sum_{i=1}^{p}\alpha_1\,\Delta y_{t-1} + \sum_{i=0}^{q}\alpha_2\,\Delta X_{t-i}^{+} + \sum_{i=0}^{q}\alpha_3\,\Delta X_{t-i}^{-}$ denotes the short-run and $\rho y_{t-1} + \varphi_1^{+}X_{t-1}^{+} + \varphi_2^{-}X_{t-1}^{-}+$ indicates the long-run estimation. In addition, $X_t^{+}$ and $X_t^{-}$ are the partial sums of POS (+) and NEG (−) changes in $X_t$. Like the ARDL bounds test, the NARDL models determine the long-run relationship between regressand or $X^{+}$, $X^{-}$, and regressors or $Y$ [55].

### 4.3. Model Stability Test

Long- or short-term cointegration does not imply a stability analysis of the results over the sample period. Therefore, the CUSUM and CUSUM of the Squares tests, the Ramsey RESET model and the Breusch–Godfrey Serial Correlation LM Q-statistics residual diagnostics test were utilized in this investigation.

### 4.4. Robustness Checks Test

The current work follows [56]'s lead in considering the robustness checks using the robustness check estimators DOLS, FMOLS, and CCR and the approaches predicated by [57]. These estimators are easy-to-use instruments for computing the cointegrating slope between integrated variables in the presence of endogenous feedback and correcting the first-order OLS bias to produce an unwelcome parameter-free asymptotic distribution. Despite the literature's availability of several robustness check approaches, these chosen estimators are significantly more reliable for resolving second-order bias, endogeneity, and serial correlation problems in cointegrating regressions [56].

## 5. Empirical Analysis and Discussion

### 5.1. Unit Root Outcomes

In this study, the authors examine the symmetrical and asymmetrical impacts of LEP on carbon dioxide emissions in Portugal, accounting for LTEC, LTRA, and LGDP from 1990 Q1 to 2020 Q4. The outcome of the stationarity test of the LS model with a breakpoint is used to gauge how well the time series variables integrate. The results of the LS with a breakpoint unit root test are shown in Table 2. These findings indicate that none of the series were stationary at level [I(0)]. However, as seen in Table 2, all variables became integrated when the first difference [I(1)] was introduced.

**Table 2.** Presents a Summary of the Outcomes of the Stationarity Test.

| | | **At Level** | | | | |
|---|---|---|---|---|---|---|
| | | LCO$_2$ | LTEC | LGDP | LEP | LTRA |
| LS | t-Statistic (tau) | −3.997259 | −4.596786 | −5.873418 | −4.589023 | −5.007489 |
| | Break Points | 2001Q4 2014Q3 | 1998Q3 2012Q3 | 1999Q1 2010Q4 | 1994Q1 2001Q3 | 1996Q4 2009Q3 |
| Test critical values | 1% level | −6.107867 | −5.831440 | −5.810780 | −5.717540 | −5.016780 |
| | 5% level | −5.495740 | −5.302100 | −5.396880 | −5.220447 | −6.311023 |
| | 10% level | −5.221680 | −4.970907 | −5.107240 | −4.961040 | −5.221680 |
| | | **At First Difference** | | | | |
| | | LCO$_2$ | LTEC | LGDP | LENG | LTRA |
| LS | t-Statistic (tau) | −6.096707 | −6.046787 | −6.18310 | −6.181579 | −6.895852 |
| | Break Points | 1999Q1 2013Q3 | 1993Q3 2003Q4 | 1996Q1 2009Q4 | 1994Q3 2014Q4 | 2007Q3 2012Q4 |
| Test critical | 1% level | −5.986360 | −6.022880 | −5.10131 | −5.824520 | −6.102211 |
| values | 5% level | −5.405120 | −5.380800 | −6.81102 | −5.297800 | −5.330001 |
| | 10% level | −5.131760 | −5.039560 | −6.20106 | −4.966920 | −5.150602 |

This current paper then moved on to estimate the BDS model of [54] following the initial assessment and using the LS model. The datasets were put through the BDS test to find any hidden stochastic nonlinear patterns (dependence and independence). The results of the BDS test are displayed in Table 3. It demonstrates that the z-statistics values are

significantly larger than the 'BDS critical values', which suggests that the selected Portugal variables are nonlinear dependent, as illustrated in Table 3.

**Table 3.** Summary of BDS Test Outcomes.

| $LCO_2$ | | | | |
|---|---|---|---|---|
| **Dimension** | **BDSStatistic** | **Std. Error** | **z-Statistic** | **Prob.** |
| 2 | 0.201424 | 0.003710 | 54.28873 | 0.0000 |
| 3 | 0.340019 | 0.005886 | 57.76742 | 0.0000 |
| 4 | 0.435383 | 0.006992 | 62.26777 | 0.0000 |
| 5 | 0.501646 | 0.007268 | 69.01935 | 0.0000 |
| 6 | 0.548595 | 0.006989 | 78.49039 | 0.0000 |
| **LTRA** | | | | |
| **Dimension** | **BDSStatistic** | **Std. Error** | **z-Statistic** | **Prob.** |
| 2 | 0.194223 | 0.008235 | 23.58503 | 0.0000 |
| 3 | 0.328173 | 0.013159 | 24.93879 | 0.0000 |
| 4 | 0.419796 | 0.015758 | 26.64054 | 0.0000 |
| 5 | 0.480806 | 0.016517 | 29.10937 | 0.0000 |
| 6 | 0.521729 | 0.016020 | 32.56717 | 0.0000 |
| **LGDP** | | | | |
| **Dimension** | **BDSStatistic** | **Std. Error** | **z-Statistic** | **Prob.** |
| 2 | 0.196699 | 0.004147 | 47.42921 | 0.0000 |
| 3 | 0.331343 | 0.006612 | 50.11263 | 0.0000 |
| 4 | 0.424008 | 0.007894 | 53.71174 | 0.0000 |
| 5 | 0.488331 | 0.008247 | 59.20983 | 0.0000 |
| 6 | 0.534142 | 0.007971 | 67.00743 | 0.0000 |
| **LTEC** | | | | |
| **Dimension** | **BDSStatistic** | **Std. Error** | **z-Statistic** | **Prob.** |
| 2 | 0.197245 | 0.004209 | 46.86240 | 0.0000 |
| 3 | 0.332030 | 0.006673 | 49.75613 | 0.0000 |
| 4 | 0.423148 | 0.007923 | 53.40840 | 0.0000 |
| 5 | 0.485615 | 0.008231 | 58.99495 | 0.0000 |
| 6 | 0.529287 | 0.007912 | 66.89886 | 0.0000 |
| **LEP** | | | | |
| **Dimension** | **BDSStatistic** | **Std. Error** | **z-Statistic** | **Prob.** |
| 2 | 0.182855 | 0.006110 | 29.92911 | 0.0000 |
| 3 | 0.301211 | 0.009720 | 30.98992 | 0.0000 |
| 4 | 0.377275 | 0.011583 | 32.57035 | 0.0000 |
| 5 | 0.424733 | 0.012082 | 35.15554 | 0.0000 |
| 6 | 0.454065 | 0.011659 | 38.94666 | 0.0000 |

*5.2. NARDL Model Estimators Outcomes*

The entire empirical findings of the "asymmetric and symmetric long-term" effects of energy productivity on environmental quality, as determined by $CO_2$ emissions in Portugal, are presented in Table 4. The analytical misspecification tests using unidirectional forward stepwise regressions showed that the nonlinear-ARDL model specification was appropriate, and the BDS test likewise showed a nonlinear dependent efficient model. In order to evaluate whether the difference between the coefficients of positive (POS) and negative (NEG) changes had the same size (symmetric effect) or a different magnitude (asymmetric effect), as indicated in Table 4, this study continued to determine whether it was statistically significant.

The regression analysis revealed that $CO_2$ emissions were decreased by both POS and NEG shocks to EP, with negative and significant coefficients at the 1% and 10% levels. As demonstrated, EP reduces $CO_2$ emissions by 3.247606% with a 1% rise in energy efficiency policy, whereas Portugal's environmental quality declines by 0.987401% with a 1% decrease

in energy efficiency policy. This suggested that using the idea of energy efficiency as a tool for policymaking would enhance Portugal's environmental quality. The increased use of biofuels in conjunction with alternative energy sources, wind power generation, technology patents, and solar system advancements are the main strategies for maintaining a low-carbon civilization in Portugal. The Portuguese authorities have consistently retained energy security as a great priority in their policy focus. However, energy issues, such as using fossil fuels in homes, commercial buildings, mechanical systems, and industries, require modern innovations to face fossil fuels' high price for living sustainably. Portugal's environmental quality would decline by 0.987401% for every 1% drop in energy productivity. This result concurs with the results of [1,38] and Hypothesis 1 in this investigation.

**Table 4.** Presents Nonlinear ARDL Long Run Form and Bounds Test Outcomes.

| Nonlinear-ARDL Long Run Form | | | | |
|---|---|---|---|---|
| **Variable** | **Coefficient** | **Std. Error** | **t-Statistic** | **Prob.** |
| LEP_POS | −3.247606 | 0.444983 | −7.298266 | 0.0000 |
| LEP_NEG | −0.987401 | 0.526886 | −1.874032 | 0.0642 |
| LGDP_POS | 0.291129 | 0.218241 | 1.333976 | 0.1857 |
| LGDP_NEG | 1.987156 | 0.905697 | 2.194064 | 0.0309 |
| LTEC_POS | 0.034088 | 0.130601 | 0.261006 | 0.7947 |
| LTEC_NEG | −0.856066 | 0.315882 | −2.710082 | 0.0081 |
| LTRA_POS | 0.717775 | 0.263122 | 2.727915 | 0.0077 |
| LTRA_NEG | −0.166010 | 0.442954 | −0.374778 | 0.7087 |
| C | 4.705552 | 0.016182 | 290.7852 | 0.0000 |
| CointEq(−1) * | −0.085673 | 0.015447 | −5.546284 | 0.0000 |
| Bounds Test | | | | |
| F-Bounds Test | | Null Hypothesis: No levels relationship | | |
| Test Statistic | Value | Signif. | I(0) | I(1) |
| | | | Asymptotic: n = 1000 | |
| F-statistic | 4.790713 | 10% | 1.85 | 2.85 |
| k | 8 | 5% | 2.11 | 3.15 |
| | | 2.5% | 2.33 | 3.42 |
| | | 1% | 2.62 | 3.77 |

Note: * represent levels that are statistically significant at the 10% level, respectively.

According to economic theory, it is still being determined how trade openness (TRA) affects carbon dioxide emissions. According to [21,58], TRA could fund innovative efforts to enhance ecologically promising projects and reduce carbon emissions. Others say that TRA will raise carbon emissions by boosting energy demand and increasing scale manufacturing. Despite inconsistent evidence on the nexus connecting TRA and carbon dioxide emissions, the results of this analysis demonstrate that the PSO and NEG shocks in TRA decreased and increased $CO_2$ emissions in Portugal, respectively. In Portugal, a 1% rise in TRA yields a 0.717775% increase in carbon dioxide emissions, which is statistically insignificant. At the same time, a 1% decrease in TRA causes a 0.166010% decrease in environmental quality. These results are supported by the empirical findings of [38,59] and Hypothesis 2 of the current investigation.

Regarding GDP growth, the findings show that volatility shock in GDP increases $CO_2$ emissions in Portugal. To put it another way, a 1% increase in GDP growth increases $CO_2$ emissions by 0.291129%; on the other hand, a 1% negative shock in GDP propels a rise in carbon dioxide emissions by 1.987156%, which was found to be statistically significant at the 5% level. In other words, an environmental quality policy that promotes green growth may improve Portugal's environmental quality. Recently, there has been a renewed emphasis on disentangling economic growth from carbon emissions. This result concurs with [60]'s study, in contrast to the studies of [45,49] and Hypothesis 3 in this paper. Developing countries have suffered negative environmental consequences due to their primary raw

material export-led growth. Further, a recent paper, [61], states that air pollutants from fossil fuel combustion are causing significant environmental challenges. Pollutant investments should be reduced by implementing smart, clean energy policies.

Regarding LTEC, POS and NEG shock LTEC have both POS and NEG impacts on environmental quality in Portugal. In particular, a 1% upsurge in investment in nonrenewable power-generating usage increases environmental degradation by 0.034088%. In comparison, a 1% decline in fossil fuel or nonrenewable energy consumption improves environmental quality by 0.856066% in real terms. These outcomes validate research from [38]. In particular, the study by [38] demonstrates that using renewable energy dramatically decreases $CO_2$ emissions in the United States. Other environmental issues, such as thermal pollution, water pollution, and solid waste disposal, must be managed, in addition to implementing the impacts of a renewable energy consumption policy in Portugal. The outcomes of the cointegration test and the ECM-based test scores are also displayed in Table 4. Further, the NARDL model outcome of 2.85 and the NARDL ECM-based test values of the ($-0.085673$) variations validated that variables are cointegrated over a long period, and that the overall effects are asymmetric in Portugal.

### 5.3. Robustness Checks Test Outcomes

Taking the robustness checks into account, Table 5 displays the DOLS, FMOLS, and CCR model that resulted when a proxy for environmental quality was used as the baseline variable for robustness testing [62–64]. The model addressed serial correlation, endogeneity issues, and second-order bias. As previously demonstrated by the NARDL results, the estimated model is statistically significant and has the expected signs. In other words, the effect of LEP on carbon dioxide emissions in Portugal is negatively significant; i.e., a 1% upsurge in LEP reduces carbon dioxide emissions by 1.686552% (DOLS), 1.415517% (CCR), and 1.430251% (FMOLS). These findings support the first hypothesis' assumption in this study and [62–64]'s studies. In the second hypothesis, this current paper assumed that the TEC significantly increased $CO_2$ emissions in Portugal during the study period. According to the robustness checks, 1% changes in the TEC increased carbon dioxide emissions by 0.168668% (DOLS), 0.138002% (CCR), and 0.145480% (FMOLS), respectively.

**Table 5.** Present Summary of Robustness Tests Outcomes.

| DOLS | | | | |
|---|---|---|---|---|
| **Variable** | **Coefficient** | **Std. Error** | **t-Statistic** | **Prob.** |
| LEP | −1.686552 | 0.206541 | −8.165718 | 0.0000 |
| LGDP | 0.479874 | 0.119618 | 4.011733 | 0.0001 |
| LTEC | 0.168668 | 0.091955 | 1.834231 | 0.0694 |
| LTRA | 0.075381 | 0.101528 | 0.742464 | 0.4594 |
| C | 0.916453 | 2.210384 | 0.414613 | 0.6793 |
| CCR | | | | |
| Variable | Coefficient | Std. Error | t-Statistic | Prob. |
| LEP | −1.415517 | 0.208118 | −6.801508 | 0.0000 |
| LGDP | 0.548103 | 0.105657 | 5.187572 | 0.0000 |
| LTEC | 0.138002 | 0.096118 | 1.435750 | 0.1539 |
| LTRA | 0.002670 | 0.097200 | 0.027466 | 0.9781 |
| C | −1.127766 | 1.933655 | −0.583230 | 0.5609 |
| FMOLS | | | | |
| Variable | Coefficient | Std. Error | t-Statistic | Prob. |
| LEP | −1.430251 | 0.206846 | −6.914557 | 0.0000 |
| LGDP | 0.560931 | 0.107820 | 5.202475 | 0.0000 |
| LTEC | 0.145480 | 0.090975 | 1.599115 | 0.1127 |
| LTRA | −0.003066 | 0.102599 | −0.029886 | 0.9762 |
| C | −1.253479 | 1.973398 | −0.635188 | 0.5266 |

In this study, the third hypothesis was proven when that TRA caused an increase in $CO_2$ emissions in Portugal during the study period. In other words, a 1% increase in the investment in fossil fuel-led traded resources increases $CO_2$ emissions by 0.075381% (DOLS) and 0.002670% (CCR). In contrast, FMOLS robustness tests revealed that the TRA model reduces carbon dioxide emissions by 0.0036%. The findings back up [65]'s paper, which found mixed results in regard to the nexus of TRA and carbon dioxide emissions. Finally, the fourth hypothesis assumed that Portugal's GDP growth rate causes a steady increase in $CO_2$ emissions, i.e., every 1% increase in GDP causes $CO_2$ emissions to rise to 0.479874% (DOLS), 0.548103 (CCR), and 0.560931 (FMOLS). This finding backs up the study outcomes of [49,58], respectively. This result shows that explanatory variables are more robust, as indicated by the R-squared value of 0.998679 and Adj. R-squared value of 0.998311, as indicated in Table 5.

### 5.4. Models Stability Results

Furthermore, the success of any econometric investigation depends on the stability of the generated model. The cumulative stability tests suggested by [66] are shown in Figures 5 and 6, and a synopsis of the residual diagnostic test for the Breusch–Godfrey Serial Correlation LM Test is shown in Table 6. These test results demonstrate that the models have no serial correlation and that the residuals follow a normal distribution. There are no issues with heteroskedasticity or misspecification, and the control variables utilized in the study have a substantial impact. Additionally, both the CUSUM and CUSUM of the Squares test indicated that the variables were stable at a significance level of 5%, as depicted in Figures 5 and 6.

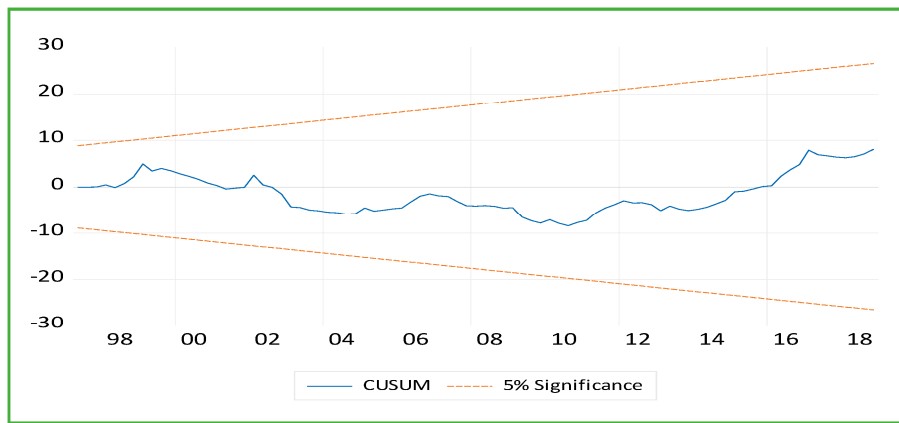

**Figure 5.** Pictorial of CUSUM Plot.

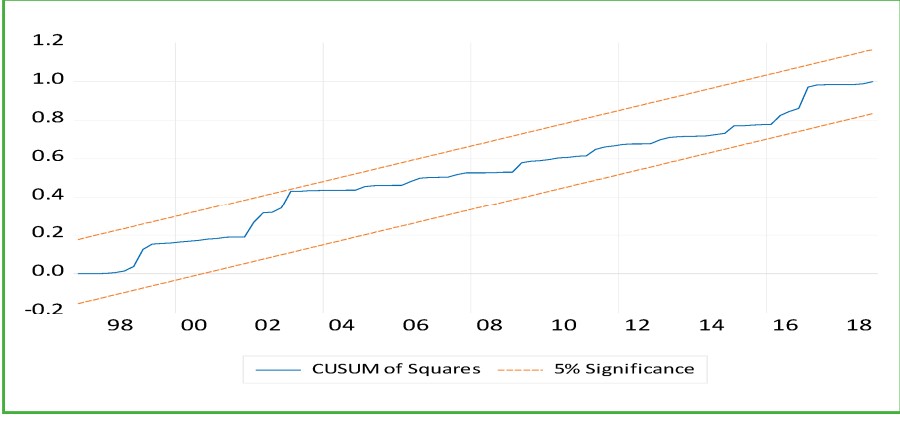

**Figure 6.** Pictorial of CUSUM of Squares Plot.

**Table 6.** LM Stability Test Outcomes.

| | | | |
|---|---|---|---|
| F-statistic | 0.956442 | Prob. F(2.86) | 0.3883 |
| Obs* R-squared | 2.458752 | Prob. Chi-Square(2) | 0.2925 |

Note: * indicate 10% level of significance.

## 6. Conclusions and Policy Implications

Since 1990, Portugal has implemented one of the most successful systems in the world for environmental governance. Despite Portugal's policy attempts to attain carbon neutrality by 2050, which includes the development of a low-carbon roadmap for industry, the nation still faces very particular environmental issues that need to be addressed. The Portuguese Government's Climate Action Plan 2021 underlines the importance of the power and utility sectors in the country's decarbonization strategy, with a strong emphasis on decarbonizing the economy and broader society through green electrification, with a particular focus on wind electricity generation. Portugal's challenge is to achieve social, economic, and environmental development that is balanced nationally and converges with other European nations. Portugal's GDP contracted by 2.9% in 2021, continuing from 2010 when the nation's economy again entered a recession. A sharp rise in energy prices, the need for the electricity grid to transition to renewable sources, and policy challenges have led to calls for robust policy intervention. In this regard, the current paper investigates the long-term asymmetric and symmetric effects of energy productivity on environmental quality in Portugal, as determined by carbon dioxide emissions. Further, the approach in this article considers other factors that affect carbon dioxide emissions, such as the total energy consumption, LGDP growth, and LTRA trade openness in the era of industry 4.0. This study used the NARDL estimators and additional robustness tests from the DOLS, FMOLS, and CCR methods to evaluate these relationships.

This study revealed that the misapplications of LTEC, LGDP, and LTRA contributed to environmental degradation in Portugal, but the robust and sound policy application of LPE improved Portugal's environment. In addition, the results verify the asymmetric impacts across the modeled variables; specifically, 1% POS and NEG shocks in energy productivity (EP) reduce carbon dioxide emissions in Portugal by 3.247606% and 0.987401 over the long term. In contrast, in terms of economic growth (GDP), POS and NEG GDP shocks contribute 0.29119% and 1.987156% to the long-term increase in carbon dioxide emissions. Further, volatility shocks to LTEC and trade openness (TRA) indicate long-term increases and decreases in $CO_2$ emissions.

Economic theory demonstrated these outcomes. Notably, a 1% increase in TEC and TRA increased $CO_2$ emissions by 0.034088% (TEC) and 0.717775% (TRA). Meanwhile, a 1% negative shock of TEC and LTRA reduce $CO_2$ emissions by 0.856066 (LTEC) and 0.166010% (LTRA), respectively. In addition, the outputs of the DOLS, FMOLS, and CCR models complement the findings of the NARDL model. According to this study, neglecting the inherent nonlinearities may result in erroneous reasoning [47,67]. Consequently, the following recommendations are presented.

This study proposes some plausible policy implications based on the estimation results. The experimental results demonstrate that energy productivity has a significant effect on reducing $CO_2$ emissions. Nonetheless, vital investments in fossil fuel resources diminish the promotion of renewable energy and expand the carbon footprint. The present analysis from this study provides valuable information that will help policymakers establish policies based on the analyzed factors (empirical results). Some policy recommendations are discussed below, based on the findings of this study:

- The Portuguese government is capitalizing on the flow of essential investment opportunities that come from developed countries; hence, to raise the level of renewable energies, the government should establish environmental standards that contribute to improving the quality of the local environment through a variety of measures represented in the use of environmentally friendly policy tools. Further, by enacting a

policy that promotes industrial waste management, financial resources from polluting sectors in the energy transition should be invested in green technologies.

- The government should create incentives to encourage green investments that contribute to the energy transition. As a result, Portugal's economic and environmental policies must be reconsidered, as it is necessary to work to achieve a balance between economic growth and emission reductions; this is particularly the case in terms of institutional investments, where it is necessary to develop encouraging regulations to promote institutional investments in renewable energy in order to increase efficiency, curb fossil energies, host polluting industries through fiscal policy and accelerate the process of implementation. Portugal's government should also take advantage of the country's energy productivity to cope with trade openness and generate opportunities to export to more developed nations that rely on environmental standards; this is in order to obtain expertise and standards that contribute to improving environmental quality. Furthermore, governments should capitalize on the observed financial inclusion by establishing financial products that contribute to the promotion of renewable energies, as demonstrated in carbon footprint reduction, by achieving a balance between the overall economic growth and environmental quality. To counterbalance the negative impact of trade openness on environmental degradation, governments should incorporate it into local, national, and regional climate change initiatives. Furthermore, the government and policymakers should increase access to green funding and promote ecologically favorable commodities to achieve carbon neutrality.

To do this, Portugal may cut tariffs and reach long-term agreements on green commodity trade. Furthermore, due to the discovery of the link between economic growth and environmental deterioration, all regions in Portugal should undertake economic development methods of eco-innovation. Finally, Portugal should accelerate its economic growth rates and reach a national income threshold level that eliminates the tradeoff between more robust economic growth and a higher carbon footprint. Based on the estimated results of this article, it is proposed that the government and policymakers establish a strategy to encourage domestic investment in alternative and cleaner energy to fulfill the Sustainable Development Goals (Goals 7, 10, and 13 are just a few examples). The administrations of Portugal should take immediate initiatives toward an energy transition. Furthermore, the government should construct additional renewable energy power plants because they are carbon-free and less expensive than natural gas or coal-fired electricity.

Even though our study has made significant contributions, some limitations remain. First, the greatest impediment to conducting this study was the need for more data from recent years, so the analysis timeframe was limited to 1990 to 2020. In terms of future research directions, the research can be expanded by incorporating other drivers of environmental quality and renewable energy into the analytical framework, such as financial development and globalization. In addition, the panel nonlinear ARDL analysis of the symmetric and asymmetric effects of economic growth, renewable energy, and foreign direct investment on industrialized and emerging countries' environmental quality could offer a robust policy perspective. Finally, other ecological quality indicators, such as capacity factors and the ecological footprint, can be studied in the future.

**Author Contributions:** Conceptualization, D.K.; methodology, D.K. and J.K.S.J.; software, J.K.S.J. and S.Y.G.; validation, R.A.C.; formal analysis, S.Y.G. and R.A.C.; investigation; J.K.S.J., resources, R.A.C.; data curation, M.A.; writing—original draft preparation, J.K.S.J. and S.Y.G.; writing—review and editing D.K., M.A. and R.A.C.; visualization, D.K.; supervision, R.A.C., G.C., M.A. and S.Y.G.; project administration, G.C. and D.K. All authors have read and agreed to the published version of the manuscript.

**Funding:** The project is funded under the program of the Minister of Science and Higher Education titled "Regional Initiative of Excellence" in 2019–2022, project number 018/RID/2018/19, the amount of funding PLN 10 788 423,16". This paper is also financed by Portuguese national funds through FCT–Fundação para a Ciência a e Tecnologia, I.P., project number UIDB/00685/2020.

**Institutional Review Board Statement:** Not applicable.

**Informed Consent Statement:** Not applicable.

**Data Availability Statement:** The variables used in this paper were collected from the database of World Bank and OECD.

**Conflicts of Interest:** The authors declare no conflict of interest.

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
