# Peer review of "The Asymmetric and Symmetric Effect of Energy Productivity on Environmental Quality in the Era of Industry 4.0: Empirical Evidence from Portugal"

_sustainability, doi:10.3390/su15054096_

Round 1
Reviewer 1 Report
The manuscript titled “The asymmetric and symmetric effect of energy productivity on environmental quality in the era of industry 4.0: empirical evidence from Portugal” aims to explore investigates the asymmetric and symmetric impact of energy efficiency on environmental quality in Portugal from 1990Q1 to 2020Q4, while accounting for the role of total energy consumption (TEC), trade openness (TRA), and economic growth (GDP) in driving environmental quality in the era of industry 4.0. The manuscript is interesting and likely to contribute to the literature in its field
Here are my minor suggestions which can improve the manuscript are reported below.
1. The manuscript must use the referencing style of Sustainability journal in the whole text
2. Line 244-247, the colour of text looks like gray , not black . this issue should be fixed
3. 2 must be subscript for the CO2 emissions. This issue should be fixed in the text.
4. The authors should explain how did they convert annual data to quarterly data for the study in the data section.
5. To the Figure 4, as a technical approach , BDS and Nonlinear ARDL bounds should be added. Because these empirical approaches implemented in the study.
Author Response
The asymmetric and symmetric effect of energy productivity on environmental quality in the era of industry 4.0: empirical evidence from Portugal
We appreciate the editor's interest in our work; it is our pleasure to thank the reviewers for providing us with their valuable feedback. The comments were constructive and beneficial, which had the potential to significantly enhance this research. Precisely, in revising the manuscript, we have made the following major changes.
In addition, the authors wish to express their gratitude to the reviewers for their thorough review of this paper. Our team agrees with all of the editorial suggestions and technical comments made by the reviewer. The paper has been revised to incorporate the comments of the reviewer. In the revised paper, the changes (additions) arising from the review are highlighted in yellow. Attached is a point-by-point response to the reviewer's concerns. Details answers are the explanations provided by the authors in response to the reviewer's comments. We are grateful for the reviewer's comments and hope he/she will find our responses satisfactory and the manuscript is now deemed suitable for publication according to the reviewer's comments.
Reviewer #1
The manuscript titled “The asymmetric and symmetric effect of energy productivity on environmental quality in the era of industry 4.0: empirical evidence from Portugal” aims to explore investigates the asymmetric and symmetric impact of energy efficiency on environmental quality in Portugal from 1990Q1 to 2020Q4, while accounting for the role of total energy consumption (TEC), trade openness (TRA), and economic growth (GDP) in driving environmental quality in the era of industry 4.0. The manuscript is interesting and likely to contribute to the literature in its field
Here are my minor suggestions which can improve the manuscript are reported below.
- The manuscript must use the referencing style of Sustainability journal in the whole text
Authors Response: the citation has been reorganized accordingly
- Line 244-247, the colour of text looks like gray not black . This issue should be fixed
Authors Response: as requested, we fixed this issue
- 2 must be subscript for the CO2 emissions. This issue should be fixed in the text.
Authors Response: as requested we fixed this issue in text and now 2 is subscript in CO2
- The authors should explain how they converted annual data to quarterly data for the study in the data section.
Authors Response: as requested we added the following sentences. “the quarterly dataset is generated from annual data using the Quadratic approach in the Eviews 12 software.”
- To the Figure 4, as a technical approach, BDS and Nonlinear ARDL bounds should be added. Because these empirical approaches implemented in the study.
Authors Response: thanks a lot for this comment, as requested, we edit Figure 4.
Reviewer 2 Report
This research paper is well written and discussed accordingly. Authors have developed the model precisely and explain the research findings systematically. Overall, the authors have suggested interesting policy implications for Portugal economy.
Author Response
Reviewer#2
This research paper is well written and discussed accordingly. Authors have developed the model precisely and explain the research findings systematically. Overall, the authors have suggested interesting policy implications for Portugal economy.
Authors Response: thanks a lot
Reviewer 3 Report
Dear authors,
It is a pleasure to review your manuscript. Improving environmental quality is the necessary response to the global climate problem, and thus to achieve sustainable development. It is a good attempt for you to study this issue from the perspective of energy productivity. The following are my comments and suggestions for your paper.
(1) In the introduction, the research objectives of this paper are not very clearly presented to readers. It even takes me a lot of time to read between the lines of your paper to find them. Also, I think you need to be more direct about the marginal contribution of your work compared to the existing literature.
(2) Your paper examines the asymmetric and symmetric effects on environmental quality using Portugal as a sample. So, what is the symmetric impact of energy productivity? And what is the asymmetric impact? Although these are explained in the form of data in your discussion section, I think the concept and distinction between the two needs to be defined beforehand.
(3) The main source of CO2 emissions is the use of various types of energy, and the increase in energy consumption naturally leads to the increase in CO2 emissions. Therefore, in my opinion, hypothesis 3 presents a well known assertion and further data validation is redundant. It is suggested that the authors differentiate between energy types to clarify which energy consumption significantly increases CO2 emissions and which energy use does not lead to CO2 emissions expansion.
(4) The policy recommendations in this paper are very poor compared to the substantial empirical discussion. The authors should provide more detailed and actionable policy implications in conjunction with the findings obtained.
Some other comments:
(5) The citation format of this paper is not in accordance with the target journal, and it is recommended to revise it.
(6) Line 369, the word "development" should be capitalized.
(7) Line 421, the equation number is the same as the previous one, which causes all subsequent equations to be incorrectly numbered.
(8) The titles of Tables 4, 5 and 6 should not be embedded in the tables.
Author Response
Reviewer #3
Dear authors,
It is a pleasure to review your manuscript. Improving environmental quality is the necessary response to the global climate problem, and thus to achieve sustainable development. It is a good attempt for you to study this issue from the perspective of energy productivity. The following are my comments and suggestions for your paper.
(1) In the introduction, the research objectives of this paper are not very clearly presented to readers. It even takes me a lot of time to read between the lines of your paper to find them. Also, I think you need to be more direct about the marginal contribution of your work compared to the existing literature.
Authors Response: as requested, the research objectives is clearly identified and study contribution has is compared with relevant existing literature
(2) Your paper examines the asymmetric and symmetric effects on environmental quality using Portugal as a sample. So, what is the symmetric impact of energy productivity? And what is the asymmetric impact? Although these are explained in the form of data in your discussion section, I think the concept and distinction between the two needs to be defined beforehand.
Authors Response: thank reviewer, as requested, the distinction between symmetric and asymmetric is clearly identified
(3) The main source of CO2 emissions is the use of various types of energy, and the increase in energy consumption naturally leads to the increase in CO2 emissions. Therefore, in my opinion, hypothesis 3 presents a well-known assertion and further data validation is redundant. It is suggested that the authors differentiate between energy types to clarify which energy consumption significantly increases CO2 emissions and which energy use does not lead to CO2 emissions expansion.
Authors Response: thank reviewer, distinction has been inserted accordingly
(4) The policy recommendations in this paper are very poor compared to the substantial empirical discussion. The authors should provide more detailed and actionable policy implications in conjunction with the findings obtained.
Authors Response: thank reviewer, as requested, the policy recommendations has been reorganized accordingly
(5) The citation format of this paper is not in accordance with the target journal, and it is recommended to revise it.
Authors Response: the citation has been reorganized accordingly
(6) Line 369, the word "development" should be capitalized.
Authors Response: thanks a lot for this comment, we edit our title
(7) Line 421, the equation number is the same as the previous one, which causes all subsequent equations to be incorrectly numbered.
Authors Response: thanks a lot for this comment, we fixed the equation numbering issue
(8) The titles of Tables 4, 5 and 6 should not be embedded in the tables.
Authors Response: thanks a lot for this comment, as requested, we edit Figures.
Round 2
Reviewer 3 Report
Dear authors.
Many thanks for resubmitting the revised version of your manuscript, which has been improved to some extent over the original version, after your efforts to revise it. However, I think the manuscript still deserves further refinement, and here are some specific comments.
1. Your response to my previous comments needs to be more detailed. In your responses, you should indicate exactly how the changes were made and where they appear in the new version, otherwise I will have reason to doubt your attitude towards the changes.
2. Regarding the point 3 of my previous comments, I do not think that your revision has met my expectations. In fact, the new hypothesis 3 is still a proposition that does not need to be verified, which is a conclusion that anyone can tell through common sense.
3. The language still needs significant improvement. It is recommended that you use a paid language editing service, such as MDPI's or another professional organization's.
Some other formatting and style problems.
4. In line 49, the citation format should be [1, 2] instead of [1], [2]. The same problem occurs in lines 53, 198, 245, 292, 321, 401, 403, 409, and 772.
5. In line 260, the citation format should be [22-24] instead of [22], [23], [24]. The same problem occurs in lines 300, 344, 380, 664.
6. In lines 270-271, the last sentence of this paragraph is semantically unclear. Also, the parentheses for the hypothetical serial number are not recommended. Same below.
7. In lines 303-304, it should be hypothesis 2.
8. In line 361, the citation format should be [7, 45-48] instead of [7], [45]-[48].
9. In line 369, what is the meaning of the sentence after the semicolon?
10. In lines 587-588, the wrong line appears in the title.
11. In line 613, the sentence needs to be better presented.
12. In line 185, the title of Figure 3 should be placed below the figure.
13. Overlap of characters appears in Tables 1, 2 and 3.
14. All tables should be kept in the same style, either open or closed.
15. All equation numbers should be right-aligned.
16. The references should be consistent in the style of all cited journal names. In contrast, in the current version there are either full journal names or abbreviations.
Author Response
Dear authors.
Many thanks for resubmitting the revised version of your manuscript, which has been improved to some extent over the original version, after your efforts to revise it. However, I think the manuscript still deserves further refinement, and here are some specific comments.
- Your response to my previous comments needs to be more detailed. In your responses, you should indicate exactly how the changes were made and where they appear in the new version, otherwise I will have reason to doubt your attitude towards the changes.
Reponse to Comment 1: We fixed the issue
- Regarding the point 3 of my previous comments, I do not think that your revision has met my expectations. In fact, the new hypothesis 3 is still a proposition that does not need to be verified, which is a conclusion that anyone can tell through common sense.
Reponse to Comment 2: We fixed the issue
- The language still needs significant improvement. It is recommended that you use a paid language editing service, such as MDPI's or another professional organization's.
Reponse to Comment 3: The manuscript is edit by professional proof-reader.
Some other formatting and style problems.
- In line 49, the citation format should be [1, 2] instead of [1], [2]. The same problem occurs in lines 53, 198, 245, 292, 321, 401, 403, 409, and 772.
Reponse to Comment 4: Thanks a lot for this suggestion
- In line 260, the citation format should be [22-24] instead of [22], [23], [24]. The same problem occurs in lines 300, 344, 380, 664.
Reponse to Comment 5: Thanks a lot for this suggestion
- In lines 270-271, the last sentence of this paragraph is semantically unclear. Also, the parentheses for the hypothetical serial number are not recommended. Same below.
Reponse to Comment 6: Thanks a lot for this suggestion. We edit sentence
- In lines 303-304, it should be hypothesis 2.
Reponse to Comment 7: Thanks a lot for this suggestion. Yes it is hypothesis 3
- In line 361, the citation format should be [7, 45-48] instead of [7], [45]-[48].
Reponse to Comment 8: Thanks a lot for this suggestion
- In line 369, what is the meaning of the sentence after the semicolon?
Reponse to Comment 9: Thanks a lot for this suggestion . We removed the sentence
- In lines 587-588, the wrong line appears in the title.
Reponse to Comment 10: Fixed
- In line 613, the sentence needs to be better presented.
Reponse to Comment 11: Fixed
- In line 185, the title of Figure 3 should be placed below the figure.
Reponse to Comment 12: Fixed
- Overlap of characters appears in Tables 1, 2 and 3.
Reponse to Comment 13: Fixed
- All tables should be kept in the same style, either open or closed.
Reponse to Comment 14: In the revised version tables are at the same style.
- All equation numbers should be right-aligned.
Reponse to Comment 15: all of them at the right-aligned in the revised version.
- The references should be consistent in the style of all cited journal names. In contrast, in the current version there are either full journal names or abbreviations.
Reponse to Comment 16: Fixed.
Round 3
Reviewer 3 Report
- Please see the attachment.

Author Response
Dear reviewer;
- The image below is your response to my last round of comments. I have asked you to clearly state in your response exactly how the changes were made and where the changes are presented in the new version. However, your response was simply "We fixed the issue". I hope you can be more detailed in your responses to reviewers' comments in the future, e.g., "We have made changes according to your comments, please see lines xx to xx in the revised version, or paragraph xx on page xx".
- This time we provided clear responses with line and page numbers
- For my comment #2, I don't see any changes in the new version (see below). For this, I think you need to provide good reasons.
- Thanks a lot for this comment regarding Hypothesis 3. We removed hypothesis 3 from our text (page 7, line 336) since it is common knowledge and already proven by plenty of studies.
- For my comment #16, There are still some references whose formatting has not been modified, e.g. as shown in the figure below. The screenshot shows just one example. Therefore, you need to check every reference carefully to make sure the style is consistent.
- In addition, we fixed the text references in the text, line 261, line 301, line 329, Line 374
- Moreover, we used software to organize references, but two journal references were problematic. We fixed the abbreviation-based journal names in the revised study, and we highlighted them in the references section.